# Fast deep reinforcement learning using online adjustments from the past

**Steven S. Hansen**<sup></sup>*, **Pablo Sprechmann**\*, **Alexander Pritzel**\*, **André Barreto**, **Charles Blundell**
{stevenhansen,psprechmann,apritzel,andrebarreto,cblundell}@google.com

DeepMind

## Abstract

We propose Ephemeral Value Adjusments (EVA): a means of allowing deep reinforcement learning agents to rapidly adapt to experience in their replay buffer. EVA shifts the value predicted by a neural network with an estimate of the value function found by planning over experience tuples from the replay buffer near the current state. EVA combines a number of recent ideas around combining episodic memory-like structures into reinforcement learning agents: slot-based storage, content-based retrieval, and memory-based planning. We show that EVA is performant on a demonstration task and Atari games.

## 1 Introduction

Complementary learning systems [McClelland et al., 1995, CLS] combine two mechanisms for learning: one, fast learning and highly adaptive but poor at generalising, the other, slow at learning and consequentially better at generalising across many examples. The need for two systems reflects the typical trade-off between the sample efficiency and the computational complexity of a learning algorithm. We argue that the majority of contemporary deep reinforcement learning systems fall into the latter category: slow, gradient-based updates combined with incremental updates from Bellman backups result in systems that are good at generalising, as evidenced by many successes [Mnih et al., 2015, Silver et al., 2016, Moravčík et al., 2017], but take many steps in an environment to achieve this feat.

RL methods are often categorised as either model-free methods or model-based RL methods [Sutton and Barto, 1998]. In practice, model-free methods are typically fast at acting time, but computationally expensive to update from experience, whilst model-based methods can be quick to update but expensive to act with (as on-the-fly planning is required). Recently there has been interest in incorporating episodic memory-like into reinforcement learning algorithms [Blundell et al., 2016a, Pritzel et al., 2017], potentially providing increases in flexibility and learning speed, driven by motivations from the neuroscience literature known as Episodic Control [Dayan and Daw, 2008, Gershman and Daw, 2017]. Episodic Control use episodic memory in lieu of a learnt model of the environment, aiming for a different computational trade-off to model-free and model-based approaches.

We will be interested in a hybrid approach, motivated by the observations of CLS [McClelland et al., 1995], where we will build an agent with two systems: one slow and general (model-free) and the other fast and adaptive (episodic control-like). Similar to previous proposals for agents, the fast, adaptive subsystem of our agent uses episodic memories to remember and later mimic previously experienced rewarding sequences of states and actions. This can be seen as a memory-based form of planning [Silver et al., 2008], in which related experiences are recalled to inform decisions. Planning

in this context can be thought as the re-evaluation of the past experience using current knowledge to improve model-free value estimates.

Critical to many approaches to deep reinforcement learning is the replay buffer [Mnih et al., 2015, Espeholt et al., 2018]. The replay buffer stores previously seen tuples of experience: state, action, reward, and next state. These stored experience tuples are then used to train a value function approximator using gradient descent. Typically one step of gradient descent on data from the replay buffer is taken per action in the environment, as (with the exception of [Barth-Maron et al., 2018]) a greater reliance on replay data leads to unstable performance. Consequently, we propose that the replay buffer may frequently contain information that could significantly improve the policy of an agent but never be fully integrated into the decision making of an agent. We posit that this happens for three reasons: (i) the slow, global gradient updates to the value function due to noisy gradients and the stability of learning dynamics, (ii) the replay buffer is of limited size and experience tuples are regularly removed (thus limiting the opportunity for gradient descent to learn from it), (iii) training from experience tuples neglects the trajectory nature of an agents experience: one tuple occurs after another and so information about the value of the next state should be quickly integrated into the value of the current state.

In this work we explore a method of allowing deep reinforcement learning agents to simultaneously: (i) learn the parameters of the value function approximation slowly, and (ii) adapt the value function quickly and locally within an episode. Adaptation of the value function is achieved by planning over previously experienced trajectories (sequences of temporally adjacent tuples) that are grounded in estimates from the value function approximation. This process provides a complementary way of estimating the value function.

Interestingly our approach requires very little modification of existing replay-based deep reinforcement learning agents: in addition to storing the current state and next state (which are typically large: full inputs to the network), we propose to also store trajectory information (pointers to successor tuples) and one layer of current hidden activations (typically much smaller than the state). Using this information our method adapts the value function prediction using memory-based rollouts of previous experience based on the hidden representation. The adjustment to the value function is not stored after it is used to take an action (thus it is ephemeral). We call our method Ephemeral Value Adjustment (EVA).

## 2 Background

The action-value function of a policy $\pi$ is defined as $Q^\pi(s, a) = \mathbb{E}_\pi \left[ \sum_t \gamma^t r_t \mid s, a \right]$ [Sutton and Barto, 1998], where $s$ and $a$ are the initial state and action respectively, $\gamma \in [0, 1]$ is a discount factor, and the expectation denotes that the $\pi$ is followed thereafter. Similarly, the value function under the policy $\pi$ at state $s$ is given by $V^\pi(s) = \mathbb{E}_\pi \left[ \sum_t \gamma^t r_t \mid s \right]$ and is simply the expected return for following policy $\pi$ starting at state $s$.

In value-based model-free reinforcement learning methods, the action-value function is represented using a function approximator. Deep Q-Network agents [Mnih et al., 2015, DQN] use Q-learning [Watkins and Dayan, 1992] to learn an action-value function $Q_\theta(s_t, a_t)$ to rank which action $a_t$ is best to take in each state $s_t$ at step $t$. $Q_\theta$ is parameterised by a convolutional neural network (CNN), with parameters collectively denoted by $\theta$, that takes a 2D pixel representation of the state $s_t$ as input, and outputs a vector containing the value of each action at that state. The agent executes an $\epsilon$-greedy policy to trade-off exploration and exploitation: with probability $\epsilon$ the agent picks an action uniformly at random, otherwise it picks the action $a_t = \arg\max_a Q(s_t, a)$.

When the agent observes a transition, DQN stores the $(s_t, a_t, r_t, s_{t+1})$ tuple in a replay buffer, the contents of which are used for training. This neural network is trained by minimizing the squared error between the network's output and the Q-learning target $y_t = r_t + \gamma \max_a \tilde{Q}(s_{t+1}, a)$, for a subset of transitions sampled at random from the replay buffer. The target network $\tilde{Q}(s_{t+1}, a)$ is an older version of the value network that is updated periodically. It was shown by Mnih et al. [2015] that both, the use of a target network and sampling uncorrelated transitions from the replay buffer, are critical for stable training.

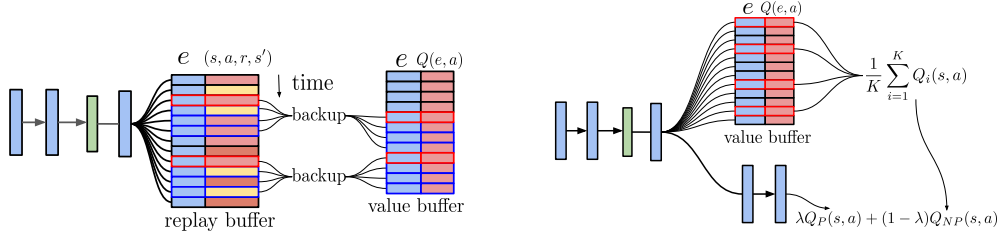

Figure 1: Left: Trajectory-centric planning over memories the replay buffer. Right: adjusting the parametric policy at action selection time using EVA.

## 3 Ephemeral Value Adjustments

Ephemeral value adjustments are a way to augment an arbitrary value-based off-policy agent. This is accomplished through a trace computation algorithm, which rapidly produces value estimates by combining previously encountered trajectories with parametric estimates. Our agent consists of three components: a standard parametric reinforcement learner with its replay buffer augmented to maintains trajectory information, a trace computation algorithm that periodically plans over subsets of data in the replay buffer, a small value buffer which stores the value estimates resulting from the planning process. The overall policy of EVA is dictated by the action-value function,

$$Q(s, a) = \lambda Q_\theta(s, a) + (1 - \lambda) Q_{\mathrm{NP}}(s, a) \tag{1}$$

$Q_\theta$ is the value estimate from the parametric model and $Q_{\mathrm{NP}}$ is the value estimate from the trace computation algorithm (non-parametric). Figure 1 (Right) shows a block diagram of the method. The parametric component of EVA consists of the standard DQN-style architecture, $Q_\theta$, a feedforward convolutional neural network: several convolution layers followed by two linear layers that ultimately produce action-value function estimates. Training is done exactly as in DQN, briefly reviewed in Section 2 and fully described in [Mnih et al., 2015].

### 3.1 Trajectory selection and planning

The second to final layer of the DQN network is used to embed the currently observed state (pixels) into a lower dimensional space. Note that similarity in this space has been optimised for action-value estimation by the parametric model. Periodically (every 20 steps in all the reported experiments), the k nearest neighbours in the global buffer are queried from the current state embedding (on the basis of their $\ell_2$ distance). Using the stored trajectory information, the 50 subsequent steps are also retrieved for each neighbour. Each of these k trajectories are passed to a trace computation algorithm (described below), and all of the resulting Q values are stored into the value buffer alongside their embedding. Figure 1 (Left) shows a diagram of this procedure. The non-parametric nature of this process means that while these estimates are less reliant on the accuracy of the parametric model, they are more relevant locally. This local buffer is meant to cache the results of the trace computation for states that are likely to be nearby the current state.

### 3.2 Computing value estimates on memory traces

By having the replay buffer maintain trajectory information, values can be propagated through time to produce trajectory-centric value estimates $Q_{\mathrm{NP}}(s, a)$. Figure 1 (Right) shows how the value buffer is used to derive the action-value estimate. There are several methods for estimating this value function, we shall describe n-step, trajectory-centric planning (TCP) and kernel-based RL (KBRL) trace computation algorithms. N-step estimates for trajectories from the replay buffer are calculated as follows,

$$V_{\mathrm{NP}}(s_t) = \begin{cases} \max_a Q_\theta(s_t, a) & \text{if } t = T \\ r_t + \gamma V_{\mathrm{NP}}(s_{t+1}) & \text{otherwise,} \end{cases} \tag{2}$$

where $T$ is the length of the trajectory and $s_t, r_{t_t}$ are the states and rewards of the trajectory. These estimates utilise information in the replay buffer that might not be consolidated into the parametric model, and thus should be complementary to the purely parametric estimates. While this process will

**Algorithm 1:** Ephemerally Value Adjustments

---

**Input** : Replay buffer $\mathcal{D}$
        Value buffer $\mathcal{L}$
        Mixing hyper-parameter $\lambda$
        Maximum roll-out hyper-parameter $\tau$

**for** $e := 1, \infty$ **do**
    **for** $t := 1, T$ **do**
        Receive observation $s_t$ from environment with embedding $h_t$
        Collect trace computed values from $k$ nearest neighbours
        $Q_{\text{NP}}(s_k, \cdot)|h(s_k) \in \text{KNN}(h(s_t), \mathcal{L})$
        $Q_{\text{EVA}}(s_t, \cdot) := \lambda Q_\theta(\hat{s}, \cdot) + (1 - \lambda)\frac{\sum_{k=0}^{K} Q_{\text{NP}}(s_k, \cdot)}{K}$
        $a_t \leftarrow \epsilon$-greedy policy based on $Q_{\text{EVA}}(s_t, \cdot)$
        Take action $a_t$, receive reward $r_{t+1}$
        Append $(s_t, a_t, r_{t+1}, h_t, e)$ to $\mathcal{D}$
        $\mathcal{T}_m := (s_{t:t+\tau}, a_{t:t+\tau}, r_{t+1:t+\tau+1}, h_{t:t+\tau}, e_{t:t+\tau})|h(s_m) \in \text{KNN}(h(s_t), \mathcal{D}))$
        $Q_{\text{NP}} \leftarrow$ using $\mathcal{T}_m$ via the TCP algorithm
        Append $(h_t, Q_{\text{NP}})$ to $\mathcal{L}$
    **end**
**end**

---

serve as a useful baseline, the n-step return just evaluates the policy defined by the sampled trajectory; only the initial parametric bootstrap involves an estimate of the optimal value function. W Ideally, the values at all time-steps should estimate the optimal value function,

$$Q(s, a) \leftarrow r(s, a) + \gamma \max_{a'} Q(s', a'). \tag{3}$$

Thus another way to estimate $Q_{\text{NP}}(s, a)$ is to apply the Bellman policy improvement operator at each time step, as shown in (3). While (2) could be applied recursively, traversing the trajectory backwards, this improvement operator requires knowing the value of the counter-factual actions. We call this trajectory-centric planning. We propose using the parametric model for these off-trajectory value estimates, constructing the complete set of action-conditional value-estimates, called this trajectory-centric planning (TCP):

$$Q_{\text{NP}}(s_t, a) = \begin{cases} r_t + \gamma V_{\text{NP}}(s_{t+1}) & \text{if } a_t = a \\ Q_\theta(s_t, a) & \text{otherwise.} \end{cases} \tag{4}$$

This allows for the same recursive application as before,

$$V_{\text{NP}}(s_t) = \begin{cases} \max_a Q_\theta(s_t, a) & \text{if } t = T \\ \max_a Q_{\text{NP}}(s_t, a) & \text{otherwise,} \end{cases} \tag{5}$$

The trajectory-centric estimates for the k nearest neighbours are then averaged with the parametric estimate on the basis of a hyper-parameter $\lambda$, as shown in Algorithm 1 and represented graphically on Figure 1 (Left). Refer to the supplementary material for a detailed algorithm.

### 3.3 From trajectory-centric to kernel-based planning

The above method may seem *ad hoc* – why trust the on-trajectory samples completely and only utilise the parametric estimates for the counter-factual actions? Why not analyse the trajectories together, rather than treating them independently? To address these concerns, we propose a generalisation of the trajectory-centric method which extends kernel-based reinforcement learning (KBRL)[Ormoneit and Sen, 2002]. KBRL is a non-parametric approach to planning with strong theoretical guarantees.[2]

For each action $a$, KBRL stores experience tuples $(s_t, r_t, s_{t+1}) \in S_a$. Since $S_a$ is finite (equal to the number of stored transitions), and these states have known transitions, we can perform value iteration

to obtain value estimates for all resultant states $s_{t+1}$ (the values of the origin states $s_t$ are not needed, as the Bellman equation only evaluates states *after* a transition). We can obtain an approximate version of the Bellman equation by using the kernel to compare all resultant states to all origin states, as shown in Equation 6. We define a similarity kernel on states (in fact, embeddings of the current state, as described above), $\kappa(s, s')$, typically a Gaussian kernel. The action-value function of KBRL is then estimated using:

$$Q_{\mathrm{NP}}(s_t, a_t) = \sum_{(s,r,s') \in S_a} \kappa(s_t, s) \left[ r + \gamma \max_{a'} Q_{\mathrm{NP}}(s', a') \right] \qquad (6)$$

In effect, the stored 'origin' states ($s \in S$) transition to some 'resultant state' ($s \in S'$) and get the stored reward. By using a similarity kernel $\kappa(x_0, x_1)$, we can map resultant states to a distribution over the origin states. This makes the state transitions from $S \to S$ instead of $S \to S'$, meaning that all transitions only involve states that have been previously encountered.

In the context of trajectory-centric planning, KBRL can be seen as an alternative way of dealing with counter-factual actions: estimate their effects using nearby transitions. Additionally, KBRL is not constrained to dealing with individual trajectories, since it treats all transitions independently.

We propose to add an absorbing pseudo-state $\hat{s}$ to KBRL's model whose similarity to the other pseudo-states is fixed, that is, $\kappa(s_t, \hat{s}) = C$ for some $C > 0$ for all $s_t$. Using this definition we can make KBRL softly blend similarity and parametric counter-factual action evaluation. This is accomplished by setting the pseudo-state's value to be equal to the parametric value function evaluated at the state under comparison: when $s_t$ is being evaluated, $Q_{\mathrm{NP}}(\hat{s}, a) \approx Q_\theta(\hat{s}, a)$ thus by setting $C$ appropriately, we can guarantee that the parametric estimates will dominate when data density is low. Note that this is in addition to the blending of value functions described in Equation 1.

KBRL can be made numerically identical to trajectory-centric planning by shrinking the kernel bandwidth (i.e., the length scale of the Gaussian kernel) and pseudo-state similarity.[3] With the appropriate values, this will result in value estimates being dominated by exact matches (on-trajectory) and parametric estimates when none are found. This reduction is of interest as KBRL is significantly more expensive than trajectory-centric planning. KBRL's computational complexity is $O(AN^2)$ and trajectory-centric planning has a complexity of $O(N)$, where $N$ is the number of stored transitions and $A$ is the cardinality of the action space. We can thus think of this parametrically augmented version of KBRL as the theoretical foundation for trajectory-centric planning. In practice, we use the TCP trace computation algorithm (Equations 4 and 5) unless otherwise noted.

## 4   Related work

There has been a lot of recent work on using memory-augmented neural networks as a function approximation for RL agents: using LSTMs [Bakker et al., 2003, Hausknecht and Stone, 2015], or more sophisticated architectures [Graves et al., 2016, Oh et al., 2016, Wayne et al., 2018]. However, the motivation behind these works is to obtain a better state representation in partially observable or non-Markovian environments, in which feed-forward models would not be appropriate. The focus of this work is on data efficiency, which is improved in a representation agnostic manner.

The main use of long term episodic memory is the replay buffer introduced by DQN.

While it is central to stable training, it also allows to significantly improve the data efficiency of the method, compare with the online counterparts that achieve stable training by having several actors [Mnih et al., 2016]. The replay frequency is hyper-parameter that has been carefully tuned in DQN. Learning cannot be sped-up by increasing the frequency of replay without harming end performance. The problem is that the network would overfit to the content of the replay buffer affecting its ability to learn a better policy. An alternative approach is prioritised experience replay [Schaul et al., 2015], which changes the data distribution used during training by biasing it toward transitions with high temporal difference error. These works use the replay buffer during training time only. Our approach aims at leveraging the replay buffer at decision time and thus is complementary to prioritisation, as it impacts the behaviour policy but not how the replay buffer is sampled from (the supplementary materials for a preliminary comparison).

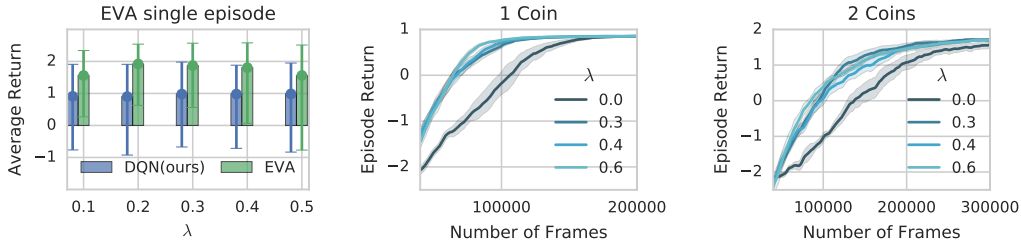

Figure 2: Left: Performance of EVA ran on a single episode using a pre-trained DQN agent (and corresponding replay buffer) for $300K$ steps and $4$ coins, see text for detailed description. Results are the average over 200 runs. Eva provides an immediate boost in performance. We can see that the benefits saturate as $\lambda$ increases. Center and Right: Performance when using EVA throughout training. $\lambda = 0$ corresponds to the DQN baseline with 1 (Center) and 2 coins (Right)

Using previous experience at decision time is closely related to non-parametric approaches for $Q$-function approximation [Santamaría et al., 1997, Munos and Moore, 1998, Gabel and Riedmiller, 2005]. Our work is particularly related to techniques following the ideas of episodic control. Blundell et al. [2016b, MFEC] recently used local regression for $Q$-function estimation using the mean of the k-nearest neighbours searched over random projections of the pixel inputs. Pritzel et al. [2017] extended this line of work with NEC, using the reward signal to learn an embedding space in which to perform the local-regression. These works showed dramatic improvements in data efficiency, specially in early stages of training. This work differs from these approaches in that, rather than using memory for local regression, memory is used as a form of local planning, which is made possible by exploiting the trajectory structure of the memories in the replay buffer. Furthermore, the memory requirements of NEC is significantly larger than that of EVA. NEC uses a large memory buffer per action in addition to a replay buffer. Our work only adds a small overhead over the standard DQN replay buffer and needs to query a single replay buffer one time every several acting steps (20 in our experiments) during training. In addition, NEC and MFEC fundamentally change the structure of the model, whereas EVA is strictly supplemental. More recent works have looked at including NEC type of architecture to aid the learning of a parametric model [Nishio and Yamane, 2018, Jain and Lindsey, 2018], sharing memory requirements with NEC.

The memory-based planning aspect of our approach also has precedent in the literature. Brea [2017] explicitly compare a local regression approach (NEC) to prioritised sweeping and find that the latter is preferable, but fail to show scalable result. Savinov et al. [2018] build a memory-based graph and plan over it, but rely on a fixed exploration policy. Xiao et al. [2018] combine MCTS planning with NEC, but relies on a built-in model of the environment.

In the context of supervised learning, several works have looked at using non-parametric type of approaches to improve the performance of models using neural networks. Kaiser et al. [2016] introduced a differentiable layer of key-value pairs that can be plugged into a neural network to help it remember rare events. Works in the context of language modelling have augmented prediction with attention over recent examples to account for the distributional shift between training and testing settings, such as neural cache [Grave et al., 2016] and pointer sentinel networks [Merity et al., 2016]. The work by Sprechmann et al. [2018] is also motivated by the CLS framework. However, they use an episodic memory to improve a parametric model in the context of supervised learning and do not consider reinforcement learning.

## 5 Experiments

### 5.1 A simple example

We begin the experimental section by showing how EVA works on a simple "gridworld" environment implemented with the pycolab game engine [Stepleton, 2017]. The task is to collect a given number of coins in the minimum number of steps possible, that can be thought as a very simple variant of the travel salesman problem. At the beginning of each episode, the agent and the coins are placed at a

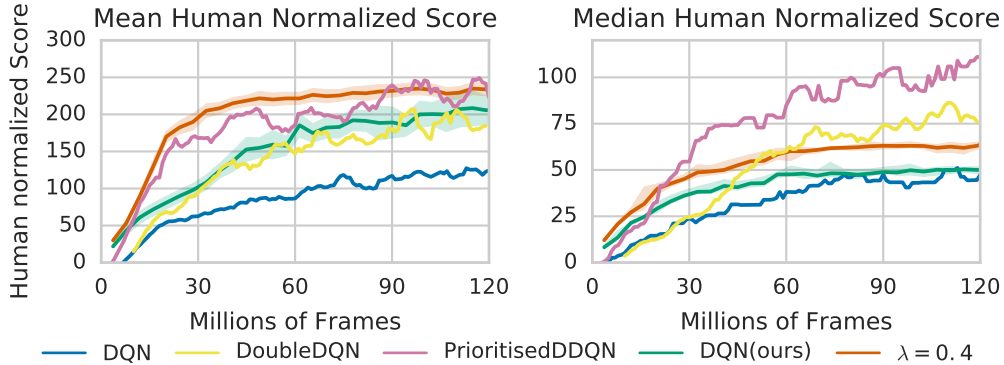

Figure 3: Comparison of the learning curves averaged over three random seeds of EVA agent the baseline according to the mean (Left) and median (Right) human normalised score. The x-axis is in billions of environment frames. We also included the original DQN results from [Mnih et al., 2015].

random location of a grid with size $5 \times 13$, see the supplementary material for a screen-shot. The agent can take four possible actions {left, right, up, down} and receives a reward of $1$ when collecting a coin and a reward of $-0.01$ at every step. If the agent takes an action that would it move into a wall, it stays at its current position. We restrict the maximum length of an episode to $500$ steps. We use an agent featuring a two-layer convolutional neural network, followed by a fully connected layer producing a 64-dimensional embedding which is then used for the look-ups in the replay buffer of size $50K$. The input is an RGB image of the maze. Results are reported in Figure 2.

**Evaluation of a single episode**   We use the same pre-trained network (with its corresponding replay buffer) and run a single episode with and without using EVA, see Figure 2 (Left). We can see that, by leveraging the trajectories in the replay buffer, EVA immediately boosts performance of the baseline. Note that the weights of the network are exactly the same in both cases. The benefits saturate around $\lambda = 0.4$, which suggests that the policy of the non-parametric component alone is unable to generalise properly.

**Evaluation of the full EVA algorithm**   Figure 2 (Center, Left) shows the performance of EVA on ful episodes using one and two coins evaluating different values of the mixing parameter $\lambda$. $\lambda = 0$ corresponds to the standard DQN baseline. We show the hyper-parameters that lead to the highest end performance of the baseline DQN. We can see that EVA provides a significant boost in data efficiency. For the single coin case, it requires slightly more than half of the data to obtain final performance and higher value of lambda is better. This is likely due to the fact that there are only $4K$ unique states, thus all states are likely to be in the replay buffer. On the two case setting, however, the number of possible states for the two coin case is approximately $195K$, which is significantly larger than the replay buffer size. Again here, performance saturates around $\lambda = 0.4$.

## 5.2   EVA and Atari games

In order to validate whether EVA leads to gains in complex domains we evaluated our approach on the Atari Learning Environment(ALE; Bellemare et al., 2013). We used the set of 55 Atari Games, please see the supplementary material for details. The hyper-parameters were tuned using a subset of 5 games (Pong, H.E.R.O., Frostbite, Ms Pacman and Qbert). The hyper-parameters shared between the baseline and EVA (e.g. learning rate) were chosen to maximise the performance of the baseline ($\lambda = 0$) on a run over $20M$ frames on the selected subset of games. The influence of these hyper-parameters on EVA and the baseline are highly correlated. Performance saturates around $\lambda = 0.4$ as in the simple example. We chose the lowest frequency that would not harm performance (20 steps), the rollout length was set to $50$ and the number of neighbours used for estimating $Q_{\text{NP}}$ was set to 5. We observed that performance decreases as the number of neighbours increases. See the supplementary material for details on all hyper-parameters used.

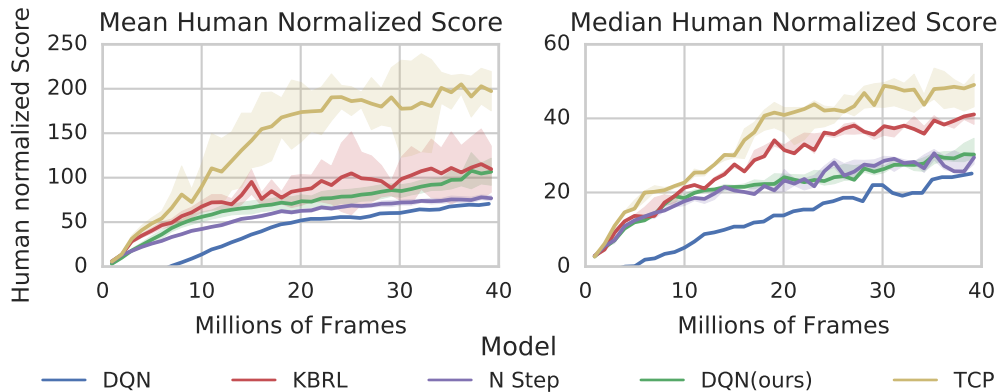

Figure 4: Comparison of the learning curves averaged over three random seeds of EVA agent with different trace computations according to the mean (Left) and median (Right) human normalised score. The x-axis is in 10s of millions of environment frames.

We compared absolute performance of agents according to human normalised score as in Mnih et al. [2015]. Figure 3 summarises the obtained results, where we ran three random seeds for $\lambda = 0$ (which is our version of DQN) and EVA with $\lambda = 0.4$. In order to obtain uncertainty estimates, we report the mean and standard deviation per time step of the curves obtained by randomly selecting one random seed per game (this is, one out of three possible seeds for each of the 55 games). For reference, we also included the original DQN results from [Mnih et al., 2015]. EVA is able to improve the learning speed as well as the final performance level using exactly the same architecture and learning parameters as our baseline. It is able to achieve the end performance of the baseline in 40 million frames.

**Effect of trace computation** To understand how EVA helps performance, we compare three different versions of the trace computation at the core of the EVA approach. The standard (trajectory-centric) trace computation can be simplified by removing the parametric evaluations of counter-factual actions. This ablation results in the n-step trace computation (as shown in 2). Since the standard trace computation can be seen as a special-case of parametrically-augmented KBRL, we also consider this trace computation. Due to the increased computation of this trace computation, these experiments are only run for 40 million frames. For parametrically-augmented KBRL, a Gaussian similarity kernel is used with a bandwidth parameter of $10^{-4}$ and a paramteric similarity of $10^{-2}$.

EVA is significantly worse than the baseline with the n-step trace computation. This can be seen as evidence for the importance of the parametric evaluation of counter-factual actions. Without this additional computation, EVA's policy is too dependant on the quality of the policy expressed in the trajectories, a negative feedback loop that results in divergence on several games. Interesting, the standard trace computation is as good as, if not better than, the much more costly KBRL method. While KBRL is capable of merging the data from the different trajectories into a global plan, it does not given on-trajectory information a privileged status without an extremely small bandwidth [4]. In near-deterministic environments like Atari, this privileged status is appropriate and acts as a strong prior, as can be seen in the lower variance of this method.

**Consolidation** EVA relies in the TCP at decision time. However, one would expect that after training, the parametric model would be able to consolidate the information available on the episodic memory and be capable of acting without relying on the planning process. We verified that annealing the value of $\lambda$ to zero over two million steps leads to no degradation in performance on our Atari experiments. Note that when $\lambda = 0$ our agent reduces to the standard DQN agent.

# 6 Discussion

Despite only changing the value function underlying the behaviour policy, EVA improves the overall rate of learning. This is due to two factors. The first is that the adjusted policy should be closer to the optimal policy by better exploiting the information in the replay data. The second is that this improved policy should fill the replay buffer with more useful data. This means that the ephemeral adjustments indirectly impact the parametric value function by changing the distribution of data that it is trained on.

During the training process, as the agent explores the environment, knowledge about value functions are extracted gradually from the interactions with the environment. Since the value-function drives the data acquisition process, the ability to quickly incorporate on highly rewarded experiences could significantly boost the sample efficiency of the learning process.

## Acknowledgments

The authors would like to thank Melissa Tan, Paul Komarek, Volodymyr Mnih, Alistair Muldal, Adrià Badia, Hado van Hasselt, Yotam Doron, Ian Osband, Daan Wierstra, Demis Hassabis, Dharshan Kumaran, Siddhant Jayakumar, Razvan Pascanu, and Oriol Vinyals. Finally, we thank the anonymous reviewers for their comments and suggestions to improve the paper.

## Footnotes

*denotes equal contribution.

[2]Convergence to a global optima assuming that underlying MDP dynamics are Lipschitz continuous, and the kernel is appropriately shrunk as a function of data.

[3] Modulo the fact that KBRL would still be able to find 'shortcuts' between or within trajectories owing to its exhaustive similarity comparisons between states

[4]To achieve this privileged status for on-trajectory information, the minimum off-trajectory similarity must be known, and typically results in bandwidth so small as to be numerically unstable

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
