[Supplementary Material · supplementary.pdf]

# Supplement: Fast deep reinforcement learning using online adjustments from the past

**Steven S. Hansen** [*], **Pablo Sprechmann** [*], **Alexander Pritzel** [*], **André Barreto**, **Charles Blundell**
{stevenhansen,psprechmann,apritzel,andrebarreto,cblundell}@google.com

DeepMind

## 1 Detailed Algorithms

---

**Algorithm 1:** Compute N-step Q-values

---

**Input** : Trajectories $\mathcal{T}_m = \{s_m^t, a_m^t, r_m^t, s_m^{t+1} | t = 0, 1, 2, ..., T-1\}$
Parametric value function $Q_\theta$
**Output :** $Q_{\mathrm{NP}}(\mathcal{T}, \cdot)$
**for** $m := 1, M$ **do**
    $s^{0:T-1}, a^{0:T-1}, r^{0:T-1} = \mathcal{T}_m$
    $V(s^{T-1}) := \max_a Q_\theta(s^{T-1}, a)$
    **for** $t := T-2, 0$ **do**
        **for** $\forall a \in A$ **do**
            **if** $a^t = a$ **then**
                $Q_{\mathrm{NP}}(s^t, a) := r^t + \gamma V(s^{t+1})$
            **else**
                $Q_{\mathrm{NP}}(s^t, a) := Q_\theta(s^t, a)$
            **end**
        **end**
        $V(s^t) := Q_{\mathrm{NP}}(s^t, a^t)$
    **end**
**end**
**return** $Q_{NP}(\cdot, \cdot)$

---

---

[*] denotes equal contribution.

---

**Algorithm 2:** Trajectory-Centric Planning

**Input** : Trajectories $\mathcal{T}_m = \{s_t^m, a_t^m, r_t^m, s_{t+1}^m | t = 0, 1, 2, ..., T-1\}$
Parametric value function $Q_\theta$

**Output**: $Q_{\text{NP}}(\mathcal{T}, \cdot)$

**for** $m := 1, M$ **do**
    $s_{0:T-1}^m, a_{0:T-1}^m, r_{0:T-1}^m = \mathcal{T}_m$
    $V(s_{T-1}^m) := \max_a Q_\theta(s_{T-1}^m, a)$
    **for** $t := T-2, 0$ **do**
        **for** $\forall a \in A$ **do**
            **if** $a_t^m = a$ **then**
                $Q_{\text{NP}}(s_t^m, a) := r_t^m + \gamma V(s_{t+1}^m)$
            **else**
                $Q_{\text{NP}}(s_t^m, a) := Q_\theta(s_t^m, a)$
            **end**
        **end**
        $V_{\text{NP}}(s_t^m) := \max_a Q_{\text{NP}}(s_t^m, a)$
    **end**
**end**
**return** $Q_{NP}(\cdot, \cdot)$

---

**Algorithm 3:** Kernel-Based Reinforcement Learning

**Input** : Query state-action pair $(\hat{s}, \hat{a})$
Replay buffer $D^a = \{s_k^a, r_k^a, s_k'^a | k = 1, 2, ..., m\} \forall a \in A$
Number of value iteration steps $limit$
Pairwise similarity function $\kappa$

**Output**: $Q(\hat{s}, \hat{a})$

**for** $k_0 := 1, m$ **do**
    **for** $a_0 := 1, A$ **do**
        **for** $k := 1, m$ **do**
            **for** $a := 1, A$ **do**
                $P(s_{k_0}'^{a_0}, a, s_k^a) := \kappa(s_{k_0}'^{a_0}, s_k^a)$
            **end**
        **end**
    **end**
**end**
$\forall a, k : V_0(s_k'^a) := 0$
**for** $i := 1, limit$ **do**
    **for** $a_0 := 1, A$ **do**
        **for** $k_0 := 1, m$ **do**
            $V_i(s_{k_0}'^{a_0}) := max_a \sum_{k=0}^m P(s_{k_0}'^{a_0}, a, s_k^a)[r_k^a + \gamma V_{i-1}(s_k'^a)]$
        **end**
    **end**
**end**
**return** $\sum_{k=0}^m \kappa(\hat{s}, s_k^{\hat{a}})[r_k + \gamma V_{limit}(s_k'^{\hat{a}})]$

## 2 Simple example

The task is to collect a given number of coins in the minimum number of steps possible, that can be thought as a very simple variant of the travel salesman problem. At the beginning of each episode, the agent and the coins are placed at a random location of a grid with size $5 \times 13$. An example of the environment with random initial location for the agent (cyan square) and the coins (yellow square) is shwon in Figure 1. The purple squares correspond to walls. The agent can take four possible actions {left, right, up, down} and receives a reward of 1 when collecting a coin and a reward of $-0.01$ at every step. If the agent takes an action that would it move into a wall, it stays at its current position. We restrict the maximum length of an episode to 500 steps.

Figure 1: Simple maze example, with two coins.

## 3 Atari Experiment Details

$$100 \times \frac{\text{Score}_{\text{agent}} - \text{Score}_{\text{random}}}{\text{Score}_{\text{human}} - \text{Score}_{\text{random}}}. \tag{1}$$

Human normalised scores are 100 for human level performance and 0 for a random agent.

For our Atari experiments, we used all the preprocessing steps used in DQN except for termination of life loss. Our DQN implementation is slightly different from the original DQN implementation. DQN runs a single environment and does one batch of replay every 4 agent steps (i.e. every 16 frames). We run 4 environments in parallel and do replay every agent step, this means that the ratio of replay to number of frames seen is roughly the same as in the original DQN implementation. In the figures in the main paper this is denoted as DQN(ours). We found this change to be beneficial in terms of runtime, as it allows us to batch the observations before passing them to the neural network. Also our evaluation procedure differs in so far that the original DQN implementation trains the agent for 1 Million frames and then evaluates the scores over 500 thousand frames to get a score, we just accumulate episode scores during training and report the average in the last training period. We found this speeds up the computation, while not majorly impacting scores. We list all our hyper-parameters in Table (1). Here 'no training period' denotes the number of frames before we start using replay. We only apply EVA once the replay buffer is fully occupied, i.e. after 500k steps (or 2M frames). We are using Adam as an optimizer with all settings being the tensorflow default, except for the learning rate. As in DQN we are using a target network, however we update every 50 steps. We found this to better for us in combination with Adam.

| | |
|---|---:|
| Temperature | 1e-05 |
| Insert period | 20 |
| $k$ | 5 |
| $\lambda$ | 0.5 |
| $M$ | 10 |
| $T$ | 50 |
| No training period | 40000 |
| Learning rate | 0.0001 |
| Replay buffer capacity | 500000 |
| Value buffer size | 2000 |
| Training batch size | 48 |
| Target network period | 50 |
| Number of parallel environments | 4 |
| Filter sizes | [8, 4, 3] |
| Filter strides | [4, 2, 1] |
| Channels | [16, 32, 32] |
| Number of fully connected activations | [256] |

Table 1: Hyperparameters used on the Atari experiments.

## 4 Additional Baselines

We want to highlight that EVA should not be seen as an alternative to variants of DQN, but rather as a strategy that could be easily combined with any of them. In fact, since EVA provides a way of exploiting the replay buffer to improve data efficiency, it can be plugged in *any* existing algorithm that uses this device. All that said, a comparative experiment is useful to provide intuition on the proposed method, so here we provide a preliminary comparison to other DQN enhancements. Figure 3 shows results for Double DQN (DDQN) and DDQN trained with prioritized replay (DDQN+PR).[2] We observe that EVA+DQN significantly outperforms DDQN in early stages of training, but achieves a lower final score. Inspired in CLS, EVA is supposed to be particularly helpful in terms of data efficiency, which we find satisfying. Although these comparisons are interesting in themselves, we emphasize that the most meaningful comparisons would be between DDQN and EVA+DDQN and between DDQN+PR and EVA+DDQN+PR. DDQN+PR achieved higher performance than either approach in isolation, and we are confident that EVA will boost the performance of both DDQN and DDQN+PR as well, as using the replay buffer to augment the behavior policy should not interfere with the modified parameter updates used by DDQN nor the skewed data distribution induced by PR. We believe that showing the complementary effects of EVA with other algorithms is a worthy pursuit for future work, but we want to emphasize that this paper is focused on conceptual clarity in presenting EVA, and so such additional experiments are out of scope.

Figure 2: Learning Curves for all atari games.

Figure 3: Performance on the Atari suite for DQN, EVA ($\lambda = 0.4$), Double DQN and double DQN with prioritized replay.

## Footnotes

[2]These curves were provided by the authors of the original papers, but unfortunately they did not provide curves for PR alone.