[Reviews · NeurIPS 2018]

Reviewer 1



This paper proposes the Ephemeral Value Adjustment method to allow deep reinforcement learning agents to simultaneously (1) learn the parameters of the value function approximation, and (2) adapt the value function quickly and locally within an episode. This is achieved by additionally storing one layer of current hidden activations. Experiments on Atari games show this method show that this method significantly improves both the learning speed and performance using the same architecture. The presentation is clear and the results are convincible. I am not familiar with this topic but the basic idea of this paper makes sense.

Reviewer 2



This paper presents a method for improving the performance of DQN by mixing the standard model-based value estimates with a locally-adapted "non-parametric" value estimate. The basic idea is to combine the main Q network with trajectories from the replay buffer that pass near the current state, and fit a value estimate that is (hopefully) more precise/accurate around the current state. Selecting actions based on improved value estimates in the current state (hopefully) leads to higher rewards than using the main Q network. I thought the idea presented in this paper was reasonable, and the empirical support provided was alright. Using standard DQN as a baseline for measuring the value of the approach is questionable. A stronger baseline should be used, e.g. Double DQN with prioritized experience replay. And maybe some distributional Q learning too. Deciding whether the proposed algorithm, in its current form, makes a useful contribution would require showing that it offers more benefits than alternate approaches to improving standard DQN. --- I have read the author rebuttal.

Reviewer 3



Summary: This paper proposes a method that can help an RL agent to rapidly adapt to experience in the replay buffer. The method is the combination of slow and general component (i.e. model-free ) and episodic memories which are fast and adaptive. An interesting part of this proposed approach is that it slightly changes the replay buffer by adding trajectory information but get a good boost in the perfoamnce. In addition, the extensive number of experiments have been conducted in order to verify the claim of this paper. Comments and Questions - This paper, in general, is well-written (especially the related works, they actually talk about the relation and difference with previous works) except for the followings: -- Section 3.1 and 3.2 do not have smooth flow. The message of these parts wasn't clear to me at first. it took me multiple iterations to find out what are the main take away from these parts. - Some variable or concepts are used without prior definition, the paper should be self-contained: 'e' in figure 1. Is it the same as 'e' in Algorithm 1? if yes, why different formats are used? if not, define. Is it a scalar? vector? - Define the counter-factual action. Again the paper should self-contained. - Line 62: 'The adjustment to the value function is not stored after it is... 'is very confusing to me. Can you clarify? - Line 104-108: 'Since trajectory information is also stored, the 50 subsequent steps are also retrieved for each neighbour.' I am not sure understood this one, can you explain? - Why in algorithm 1, 'h_t' is appended to the replay buffer but it is not sth that was mentioned in the text, eqautions, or figure. Can you explain? In general, I like the idea of this paper and I wish the text was more clear (refer to above comments) Minor: -Line 123: 'W Idealy' is a typo? - In Algorithm 1: ' ...via the TCP algorithm', referring to the TCP equation(s) would be a better idea